# Population Structure and Genetic Diversity of Italian Beef Breeds as a Tool for Planning Conservation and Selection Strategies

**DOI:** 10.3390/ani9110880

**Published:** 2019-10-29

**Authors:** Maria Chiara Fabbri, Marcos Paulo Gonçalves de Rezende, Christos Dadousis, Stefano Biffani, Riccardo Negrini, Paulo Luiz Souza Carneiro, Riccardo Bozzi

**Affiliations:** 1Dipartimento di Scienze e Tecnologie Agrarie, Alimentari, Ambientali e Forestali, Università di Firenze, 50144 Firenze, Italy; christos.dadousis@unifi.it (C.D.); riccardo.bozzi@unifi.it (R.B.); 2Associazione Nazionale Allevatori Bovini di Razza Piemontese, 12061 Carrù, Italy; mpgrezende@gmail.com; 3Consiglio Nazionale delle Ricerche, Istituto di Biologia e Biotecnologia Agraria, 20133 Milano, Italy; 4Associazione Italiana Allevatori, 00161 Roma, Italy; riccardo.negrini@unicatt.it; 5Istituto di Zootecnica, Facoltà di Scienze Agrarie, Alimentari e Ambientali, Università Cattolica del S.Cuore, 29100 Piacenza, Italy; 6Universidade Estadual Sudoeste da Bahia, Jequié, Bahia 45205-490, Brazil; carneiropls@gmail.com

**Keywords:** genetic diversity, beef cattle, pedigree analysis, autochthonous breeds, conservation

## Abstract

**Simple Summary:**

The recent alarming reports on global climate change and the challenges facing the agricultural sector to meet the increase in meat consumption, impose research in biodiversity. An important genetic pool of local breeds might play a crucial role in the near future to address these challenges. Although Italy is considered as one of the richest countries in biodiversity, there are autochthonous cattle breeds under extinction. To safeguard biodiversity and increase genetic diversity within breeds, appropriate management tools must be developed. To achieve this, precise knowledge of the population structure and genetic diversity per breed are required. This study analyzed pedigree data of six local beef breeds: Calvana, Mucca Pisana, and Pontremolese (from the region of Tuscany), all under extinction, and Sarda, Sardo Bruna, and Sardo Modicana, from the island of Sardinia, that are larger in number but of lower productivity. In addition, the study investigated the population structure of the cosmopolitan beef breeds, Charolais and Limousine, reared in the same regions and undergoing selection. The high mating percentage between relatives for Mucca Pisana and Calvana is an alarming situation for these breeds. The population structure of the Sardinian breeds suggests the application of breeding programs.

**Abstract:**

The aim was to investigate the population structure of eight beef breeds: three local Tuscan breeds under extinction, Calvana (CAL), Mucca Pisana (MUP), and Pontremolese (PON); three local unselected breeds reared in Sardinia, Sarda (SAR), Sardo Bruna (SAB), and Sardo Modicana (SAM); and two cosmopolitan breeds, Charolais (CHA) and Limousine (LIM), reared in the same regions. An effective population size ranges between 14.62 (PON) to 39.79 (SAM) in local breeds, 90.29 for CHA, and 135.65 for LIM. The average inbreeding coefficients were higher in Tuscan breeds (7.25%, 5.10%, and 3.64% for MUP, CAL, and PON, respectively) compared to the Sardinian breeds (1.23%, 1.66%, and 1.90% in SAB, SAM, and SAR, respectively), while for CHA and LIM they were <1%. The highest rates of mating between half-siblings were observed for CAL and MUP (~9% and 6.5%, respectively), while the highest rate of parent–offspring mating was ~8% for MUP. Our findings describe the urgent situation of the three Tuscan breeds and support the application of conservation measures and/or the development of breeding programs. Development of breeding strategies is suggested for the Sardinian breeds.

## 1. Introduction

Cattle domestication started in Southwest Asia, in the 9th millennium BC [1,2], while in Europe it began between 8800 to 8000 BC [3], due to migration from the Near East. The effective size of female cattle founders has been estimated to be ~80 animals [2,3]. Over time, natural and artificial selection resulted in the development of various breeds in the world. Artificial selection became more intense due to the Industrial Revolution and urbanization which begun in the 19th century and drastically changed global food consumption as well as increased the request for meat production [4]. This further led to the development of modern breeding both in plants and animals at the beginning of the 20th century [5] to meet the levels of increased food consumption. For several decades breeding programs were mainly focused on the development of high-performance cattle breeds, specialized for dairy, beef, or dual-purpose (milk and meat). As a result, today’s global food market is heavily based on cosmopolitan breeds. Nevertheless, the high success of cosmopolitan breeds worldwide resulted in the loss of interest in local breeds, which represented valuable genetic resources [6,7,8]. This situation is very likely to continue in the future based on predictions for the upcoming decades of a continued increase in the global population and food consumption [5,9,10,11].

The Food and Agricultural Organization (FAO) has reported 1224 local cattle breeds worldwide [12]. From those, 181 are extinct, 105 at critical risk, and 140 are considered endangered. The majority of those breeds are of European and Caucasian origin (119 extinct, 91 at critical risk, and 108 endangered). In Italy, 61 cattle breeds are registered, 51 of which represent local breeds. From those, 18 are extinct, 7 at critical risk, 8 endangered, and 3 in a vulnerable situation [13]. The Italian Breeders Association (AIA; Associazione Italiana Allevatori, Rome), has officially recognized 16 local cattle breeds at risk of extinction. From those, six are considered as beef (Calvana, Mucca Pisana, Pontremolese, Sarda, Sardo Bruna, and Sardo Modicana) and the remaining 10 as dual-purpose breeds. All 16 breeds are enrolled in the register of autochthonous cattle populations at limited diffusion (Registro Anagrafico delle razze bovine autoctone a limitata diffusione), with the aim to safeguard and adopt conservation measures for them. There is no breeding program running for any of these breeds and some populations are under extinction or at critical numbers, with a high risk of extinction. Regarding the Italian beef sector, the preference of Italian farmers for the Piedmontese, firstly, and later on for the imported Limousine and Charolais, over other beef breeds has resulted in a loss of interest for the rest of the Italian local beef breeds in the past several decades. 

Nevertheless, there exist advantages related to the presence and maintenance of local breeds [14]. Firstly, local breeds represent a significant genetic and economic resource, being able to adapt to various landscapes where cosmopolitan breeds cannot benefit. They are more rustic and resistant to their local environment than their cosmopolitan counterparts. Moreover, they represent an important gene bank that could be essential to address future climate changes, or potential disease outbreaks [15], and hence to preserve the global food production chain [16]. In addition, they play an important role in the preservation of human cultural inheritance. For example, local breeds have been used by farmers for organic farming and manufacturing of niche products in mountainous regions. Nevertheless, low production remains the limiting factor for the farming of local breeds that endangers their existence.

Genetic diversity is a primary step for the establishment of a breeding program or to take conservation measures. It is defined as the measure of genetic differences between and within groups of animals and is highly related to the selection and adaptation of a breed to the local environment. There are several causes that influence genetic diversity, such as migration, mutation, selection, drift, bottleneck, and inbreeding [16]. The first two processes may also bring an increase in genetic diversity, while selection via assortative mating and high levels of inbreeding (as a result of unsupervised mating among relatives) can lead to allele fixation. The inbreeding rate (Δ*F*; per year or generation) has been used to evaluate how genetic diversity evolves during breed history [17]. In particular, Δ*F* summarizes the increase in inbreeding values per generation (or year) at a population level, providing an overview of genetic diversity and the risk of inbreeding depression. This last phenomenon is essential to be controlled in livestock species, because it can cause a decrease in performance (such as growth, meat quality, and quantity) [18,19] and reduced fitness [17]. The rate of inbreeding is also related to the effective population size (*Ne*) by the equation Δ*F* = 1/2*Ne*. *Ne* is defined as the number of individuals that effectively participate in producing the next generation and is often lower than the census of the total number of individuals [20]. Consequently, it is a factor that contributes to the genetic diversity of a breed and its conservation status.

Pedigree analysis is a primary step that enables the characterization of the genetic diversity of populations: it identifies genetic variability and changes in the population structure in consecutive generations. It describes and quantifies the increase of homozygosity and the level of inbreeding in the population, important factors to be considered either for breeding or conservation schemes. Indeed, demographic analyses can also help to understand factors regarding genetic history, conservation status of a population, and relationships within and between breeds [21]. Although several studies have been carried out on the population structure and genetic diversity of cattle [22,23,24], only a few were focused on small populations. 

Our aim was to investigate the population structure and genetic diversity of six Italian local beef breeds, three reared in Tuscany (Calvana (CAL), Mucca Pisana (MUP), Pontremolese (PON)) and three in Sardinia (Sarda (SAR), Sardo Bruna (SAB), and Sardo Modicana (SAM)), utilizing pedigree information to support the development of strategies for conservation or breeding. Following FAO legislation, CAL, MUP, and SAM were classified as endangered breeds, while PON was set to a critical situation. No risk of extinction exists for SAR and SAB breeds, with each of these two breeds having a few thousand animals. In addition, to compare with cosmopolitan breeds, we analyzed pedigree data from the Italian populations of Charolais (CHA) and Limousine (LIM). 

## 2. Materials and Methods 

### 2.1. Data

Pedigree information of the local breeds was supplied by the Italian Breeders Association (AIA; Associazione Italiana Allevatori, Rome) and by the breeding association (Associazione Nazionale Allevatori delle razze bovine Charolaise e Limousine–ANACLI, Rome) for the two cosmopolitan breeds. Our full dataset included animals born between 1980 and 2018, and consisted of 2798 CAL, 3399 MUP, 328 PON, 97,163 SAR, 74,981 SAB, 25,355 SAM, 99,464 CHA, and 322,321 LIM cattle. Table 1 summarizes the pedigree data per breed.

### 2.2. Pedigree Analysis

Genetic analysis was carried out with the ENDOG v4.8 software [25]. Pedigree completeness was evaluated with the following parameters: (i) equivalent complete generations (equiGen; defined as the sum for (1/2)^n^, where n is the number of generations separating the individual from each of its known ancestors [26]); (ii) maximum complete generations (maxGen; number of generations separating an animal from its furthest ancestor [27]); and (iii) full complete generations (fullGen; number of generations separating the offspring from the furthest generation, where both parental lines of the individual are known. Ancestors with both parents unknown were considered as founders). To identify individuals with insufficient pedigree information to estimate inbreeding, the pedigree completeness index (PCI) was also calculated [28].
$$PCI=\frac{4\times C_{sire}\times C_{dam}}{C_{sire}+C_{dam}}$$
where *C_sire_* and *C_dam_* were contributions from the paternal and maternal lines calculated (separately for each line) as
$$C=\frac{1}{d}\overset{d}{\underset{i=1}{\sum }}g_{i}$$
where *g_i_* is the proportion of known ancestors in generation *i* and *d* is the total number of generations. Generation interval (GI) was defined with two measures: (i) the average age of parents at the birth of all their progenies and (ii) the average age of parents at the birth of the progenies that were used for reproduction. Both parameters were calculated for the classical four pathways (father–son, father–daughter, mother–son, and mother–daughter).

### 2.3. Genetic Diversity

Genetic diversity was described with three parameters: (i) inbreeding coefficient (*F*; the probability that an individual has two identical alleles by descent) calculated according to Meuwissen and Luo [29]; (ii) average relatedness coefficient (AR; the probability that an allele randomly chosen from the whole population belongs to a given animal) that defines the mean relationship of each individual with the rest of the population, and was computed following Gutiérrez et al. [25]; and (iii) rate of inbreeding ($\Delta F=\frac{1}{2*Ne}$), where *Ne* is the effective population size. *Ne* was computed via regression as following:$$Ne=\frac{1}{2*b}$$
where *b* is the regression coefficient of the individual *F* over equiGen. To overcome pedigree incompleteness, equiGen was used.

In addition, for a better description of the population structure within each breed, the frequency of mating between close relatives—full-siblings (sibs), half-sibs, and parent with offspring—were calculated.

Also, the effective number of founders (*fe*) [22], and the effective number of ancestors (*fa*) [25] were considered. The *fe*/*fa* ratio and the number of ancestors explaining 50% [22,24] of the genetic contribution (ANC_50), expressed as a percentage on reference population [25], were also calculated. Ratio *fe*/*fa* = 1 shows the absence of bottleneck in the population under study, and low ANC_50 is an indicator of the founder effect [30]. 

Pedigree content (i.e., the proportion of known parents in each generation) was analyzed for each breed to estimate the contribution of each ancestor (for male and female lines) up to the fifth parental generation [25].

### 2.4. Principal Component Analysis

Principal component analysis (PCA) summarizes correlated variables into a reduced set of mutually uncorrelated variables (PCs, principal components), allowing a dimensionally reduced visualization while keeping a certain amount of the original variance. Each of the PCs contains all the original variables. The PCs are constructed by maximum variability explained in the data and with the constrain to be orthogonal to each other. This helps to summarize information and to better study relationships among the samples [31]. PCA was performed on a set of population parameters to identify potential differences among the breeds under study. The parameters considered were average inbreeding coefficient (AVG_F), true mean inbreeding (TMI; including only the animals with at least three complete generations traced), GI, ANC_50, average relatedness (AR), average pedigree content (P_CONT), and *Ne* calculated based on equiGen. Moreover, the relative change of population size, expressed both as average population size ratio ($\mathrm{APSR}=mean\left[\left(\frac{N_{y}}{Ny+1}\times 100\right)-100\right]$, where *N* is the total number of animals per year and *y* = {2000, …, 2016}), and average standard deviation of population size ratio (APSSD), were included. Past software was used for the PCA [32]. 

## 3. Results

### 3.1. Pedigree Analysis

The pattern of male to female ratio per generation was similar for CAL, MUP, SAM, SAR, CHA, and LIM, with more stable numbers between males and female during the generations, while SAB and PON had fluctuating trends (Appendix A). Table 2 summarizes the average of the four pedigree completeness parameters for each breed. 

MUP had the highest PCI values (74%), followed by CAL and LIM (66% and 59%, respectively). Intermediate PCI values were observed for CHA, PON, and SAM, while SAR and SAB had the lowest PCI values. As expected, pedigree quality increased over time with a similar pattern for both PCI and equiGen indices (Appendix A). GI in years was rather high for all local breeds, with the lowest values observed in SAM (7.8) and the highest in SAB (13.3). SAR and CAL had similar GI (~10). As shown in Table 3, LIM and CHA had a lower average GI (7.0 and 6.7, respectively).

It should be noted, however, that there was variation within each breed, with standard deviation (SD) estimates of GI being equal to the mean or slightly higher. Moreover, the highest GI for the father–son and father–daughter paths was found for PON. The lowest values for the father–son path were observed for SAB and CHA, while CHA also had the lowest GI for the father–daughter path. The maternal intervals were shorter than the paternal for MUP and PON, while for SAB, SAM, CHA, and LIM, maternal pathways were greater. Equal father/mother to daughter GI was observed for SAR (10.7). CAL showed similar GI in the four pathways which varied between 9.55 (father–son) and 10.54 (mother–son) (Table 3). The largest differences between maternal and paternal pathways were observed for PON (paternal pathway ~15 years, maternal pathways ~9 years) as shown in Appendix A.

### 3.2. Genetic Diversity

Estimated genetic parameters per breed are summarized in Table 4. The average *F* for the Tuscan breeds were 7.25%, 5.10%, and 3.64% for MUP, CAL, and PON, respectively. Sardinian breeds showed lower values (1.23%, 1.66%, and 1.90% for SAB, SAM, and SAR, respectively), while CHA and LIM had inbreeding coefficients less than 1%. True mean inbreeding (TMI) was higher than *F* values in all the breeds studied.

In general, inbreeding increased by generation. For CHA and LIM, the increase was relatively small (Figure 1).

Estimates of *Ne* also varied among breeds. For the local breeds, the values ranged from 14.62 (PON) to 39.79 (SAM). *Ne* was similar for MUP, SAB, SAR, and CAL. Concerning the two cosmopolitan breeds, *Ne* estimates were much higher (Table 4). The average Δ*F* was very low in LIM, CHA, and SAM, while the Tuscan breeds, SAR and SAB, had inbreeding rates ranging between 2.54% (CAL) and 3.42% (PON). The AR values, expressed as a percentage, were generally higher in breeds with high *F*. The *fe*/*fa* ratio was practically around 1 for all the local breeds indicating the absence of narrow bottlenecks, whereas CHA and LIM had a higher ratio (3 and 2.1, respectively) (Table 5). 

The estimated ANC_50 reflects the size of the different populations, with extremely low values found for the Tuscan breeds, intermediate for SAM (~100), and high values (>200) for SAB, SAM, CHA, and LIM (Table 5). APSR and APSSD were much higher (>|100|) in PON, compared to the rest, indicating large fluctuations in the population size throughout the years. For the rest of the breeds, APSR varied between −4.38 (SAM) and 6.04 (LIM), while APSSD ranged between 6.25 and 27.57 (for LIM and MUP, respectively).

Regarding P_CONT, CAL and MUP breeds presented the highest average values of the first generation (92% and 94%, respectively), followed by LIM, whereas the lowest value was found for SAB (48%) (Table 5). The percentages of known parents, grandparents, great grandparents, and so on, for both sire and dam lines are expressed in Appendix A. CAL, CHA, and LIM had more complete paternal than maternal lines, while the opposite was found in PON. Incomplete pedigree was found for all the Sardinian breeds, even in recent generations. In contrast MUP had almost complete information up to the fifth generation.

All breeds had a very low percentage of mating between full-sibs (<1%) with the highest value being observed for PON (0.6%) (Figure 2). 

However, clear differences were observed for the half-sibs and the parent–offspring matings. In these cases, CAL and MUP had the highest percentages (9.25% and 6.65%, respectively) for the matings between half-sibs; for parent–offspring percentages were 8.2% and 6.25% for MUP and CAL, respectively. The cosmopolitan breeds had the lowest values in all cases. 

### 3.3. PCA of the Population Structure Parameters

The biplot of the first two PCs, explaining together 78.31% of the total variation among breeds, is shown in Figure 3. In general, PC1 (capturing 50.27% of the total variability) separated the local from the cosmopolitan breeds, with the exception of SAM which was placed closer to the cosmopolitan breeds, while PC2 (explaining 28.04% of the variability) further separated the Tuscan from the Sardinian breeds. More precisely, three groups were formed: (i) CAL and MUP; (ii) SAR and SAB; and (iii) LIM and CHA. SAM clustered between the Sardinian and the cosmopolitan breeds, while PON was on the sideline. CAL and MUP were located near the parameters linked to inbreeding and relatedness (i.e., AVG_F, AR, and TMI), while CHA, LIM, and SAM were connected to the effective population size (*Ne*) and the *fe*/*fa* ratio. In PC1 all parameters were loaded except P_CONT, while PC2 was mainly described by the P_CONT and AVG_F, with *Ne* and *fe*/*fa* ratio having loadings close to zero.

## 4. Discussion

To the best of our knowledge, this is the first study utilizing full pedigree records for the CAL, MUP, PON, SAR, SAB, and SAM local beef breeds together with two Italian beef populations of the CHA and LIM. Merging those breeds, our dataset consisted of three subgroups: three Tuscan breeds under extinction (CAL, MUP, and PON); three local breeds from the island of Sardinia (SAR, SAB, and SAM, each consisting of a large population and without undergoing a breeding program); and two cosmopolitan beef breeds (CHA, LIM), that are mainly reared in Italy in the regions of Tuscany and Sardinia and have recently set up a national breeding scheme. 

The three Tuscan breeds are under extinction, hence drastic measures need to be taken for their conservation. At present, only 263 CAL (37 males and 226 females), 346 MUP (52 males and 294 females), and 52 PON (8 males and 44 females) cattle are alive. A pedigree analysis to investigate the relationships among individuals and the levels of inbreeding within each breed is a primary step. 

### 4.1. Pedigree Analysis

The level of inbreeding within a breed is closely related and dependent on the pedigree completeness [33], because incomplete pedigree data can underestimate the level of inbreeding in a population [34]. In general, the degree of pedigree completeness was lower in Sardinian than in Tuscan and cosmopolitan breeds (Table 2). Cappelloni [35] reported pedigree completeness of 2.44, 3.18, and 1.72 equiGen for CAL, MUP, and PON, respectively, which is lower than those found in our analysis. This was somehow expected, since the quality of the pedigree data has increased over time and in more recent years. Pedigree completeness of all the local breeds in our study was also higher compared to Spanish local beef breeds investigated by Gutiérrez et al. [36], but similar to the more recent study by Cañas-Álvarez et al. [37], who focused on the same Spanish populations analyzing demographical changes until 2009. Torrecillas et al. [38] also analyzed MUP pedigree data, but equiGen values were lower than in the current study (2.26). This could be attributed to the smaller number of animals in the pedigree (*n* = 1231) as well as to higher pedigree incompleteness.

Regarding the cosmopolitan breeds, the number of equiGen found in LIM was similar to Slovenian Limousine (3.38) reported by Kadlečík et al. [27]. In general, LIM and CHA had lower values of pedigree completeness compared to previously reported data on European Charolais (ranging from 8.3 in Swedish populations to 9.3 in French populations) and European Limousine (6.5 in Irish population to 7.5 in Swedish and British populations) [22]. However, values of equiGen reported in these studies were averaged over a specific time period (e.g., between 2004–2008 in European Charolais).

Gutiérrez et al. [36] and Cañas-Álvarez et al. [37] analyzed Spanish local beef breeds; their average GI in years was smaller (from 3.75 in Sayaguesa to 7.83 in Morucha) than those found in our study. The GI values of LIM and CHA were comparable to other commercial breeds like Angus and Nellore [39]. For PON, MUP, and SAR our analysis showed the longest GI of the sire–offspring pathways compared with the dam–offspring pathways (Table 3). Similar results have been reported by Mc Parland et al. [40] in Charolais, Limousine, Hereford, Angus, Simmental, and Holstein Friesian breeds. For the two cosmopolitan breeds and SAB results were opposite. Similar findings, however, have been previously reported in Illawarra Shorthorn, Hereford, and two Asturiana breeds [41,42,43]. This could be partly attributed to the use of artificial insemination in cosmopolitan breeds nowadays, which is almost entirely missing in the local breeds, and farming practices (e.g., the longevity of the dams within each breed). 

### 4.2. Genetic Diversity

The highest average inbreeding values were observed for Tuscan breeds (from 3.64% for PON to 7.25% for MUP) as shown in Table 4. The inbreeding level of CHA (0.96%) was similar to Swedish, Irish, and Danish Charolais, but higher than French Charolais (0.67%) [22]. The LIM had lower inbreeding than American (1%) [44] and Irish (1.08%) Limousine [40].

The TMI values were higher in breeds with a PCI lower than 40% (PON, SAR, SAB, and SAM) (Table 2 and Table 4), confirming the underestimation of the inbreeding coefficient when pedigree is incomplete [30]. The level of inbreeding had, in general, a linear increase in local breeds among generations (Figure 1), except for PON where the changes in the number of animals throughout generations produced an erratic trend.

Another measure commonly used to assess the genetic variability within a breed is *Ne*. Meuwissen [45] proposed a threshold value of 50 animals to prevent the loss of genetic variability. The local breeds in our study had lower values of *Ne* than the threshold proposed, ranging between 14 and 40. However, a common problem related to the analysis of *Ne* is the amount of missing data in the pedigree. Boichard et al. [30] argued that the low pedigree completeness could result in overestimation of the *Ne*. In our study, the Sardinian breeds had very low pedigree quality, as depicted by the estimation of equiGen and the PCI (Table 4), and although the population size of those breeds is large enough, *Ne* was low (SAR = 16.64; SAB = 18.91; SAM = 39.79), suggesting that the situation could be more alarming in terms of loss of genetic variability. 

The situation is different for the Tuscan breeds. The higher pedigree quality (greater values of equiGen and PCI) allowed for more accurate estimation of *Ne*. The low values found for the three populations, ranging between 14.6 (PON) and 19.7 (CAL), together with the small populations and the number of farms, report these breeds to be in an alarming situation. Notably, nowadays ~48% of the alive CAL cattle and ~70% of PON belong to four farms, while for MUP, five farms keep ~79% of the total population. This is a worrisome fact. In the case of an outbreak disease in the area there will be a thread on the existence of the Tuscan breeds.

Compared to other Italian local beef breeds (Chianina, Marchigiana, and Romagnola) analyzed by Bozzi et al. [46], the three Tuscan breeds had lower *Ne* estimates. In a more recent study, Mastrangelo et al. [21] analyzed genomic data for the same local and cosmopolitan breeds presented in this study. The reported *Ne* estimates differed from our estimates (Calvana = 33.5; Mucca Pisana = 8.7; Pontremolese = 7.2; Sarda = 62.2; Sardo Bruna = 1021.3; Sardo Modicana = 54.8; Charolais = 67.8; Limousine = 468.9); however, this discrepancy could be attributed to a different approach used from the authors, who estimated *Ne* from the relationship between linkage disequilibrium (LD), *Ne,* and recombination rate. Moreover, their analysis was based on a small sample of the total population per breed (24, 23, 24, 30, 10, 28, 25, and 20, for CAL, MUP, PON, SAR, SAB, SAM, CHA, and LIM, respectively), which might not be representative of the population. Future genomic analysis utilizing a larger number of animals could help in reducing this discrepancy. Nevertheless, this is another indicator of the sensitivity of the *Ne* estimates upon the methodology applied.

Regarding the level of inbreeding, FAO suggests a threshold of 1% per generation to maintain reproductive fitness [47]. Several studies had analyzed Δ*F* with different approaches, estimating annual Δ*F* [22,36] or per generation [48], while others emphasize Δ*F* during the last generation [46]. The six local breeds had Δ*F* greater than 1% per generation with the highest value observed for PON (3.42%). The AR between individuals of the Tuscan populations ranged from 6.4 to 10.5, indicating that animals shared a high percentage of alleles in relation to the population. The AR values observed in Sardinian breeds were low (<0.4%), but this could also be an artefact due to the lack of complete pedigree data.

The *fe*/*fa* ratio close to 1 that was found in all local breeds indicates a high balance between the founders’ contributions and consequently, an absence of bottleneck effect [30]. As expected, LIM and CHA presented greater values, which could be mainly attributed to selection. Estimates of ANC_50 (Table 5) suggested the presence of founder effect for the Tuscan breeds. Regarding the pedigree content, in general, the completeness of sire pathways was higher in more distant generations. Similar findings have been reported in Spanish local beef breeds [36]. The maternal line information was more complete only in the last generations.

The proportions of mating between close relatives were also examined. The most alarming situations were observed for CAL, PON, MUP, and SAR, with rates of half-sibs matings >4% and up to 9.25% for CAL (Figure 2). Although matings between full-sibs are avoided, high rates of mating between half-sibs and parent–offspring are worrisome. In contrast, the cosmopolitan breeds had a very low percentage of matings between close relatives. This was to some extent expected, since both populations in Italy were based so far on imported animals and semen from abroad, while the national breeding program was recently initialized. Finally, a PCA performed on a set of estimated population parameters revealed the similarity and a common structure between CHA–LIM, SAR–SAB, and CAL–MUP (Figure 3).

### 4.3. Measures of Conservation

Present results suggest the necessity of safeguarding measures that will guarantee the physical, economic, and logistic viability of the three Tuscan breeds and thereby their existence. High rates of parent–offspring and half-sibs matings in MUP outline the necessity for the development of an appropriate mating scheme. Several ways exist to preserve animal populations from extinction and maintain genetic diversity within a population: (i) genetic/genomic tools, (ii) biotechnology, (iii) management, (iv) scientific support, (v) cultural relatedness, and (vi) political engagement. In the first two categories, cryoconservation of semen, embryos and oocytes could be reported [4,49]. In addition, control of mating targeting either sustaining biodiversity or maintaining favorable characteristics of the animals, or both, could be applied [50,51]. Optimal contribution offers one possible way to achieve this [52]. Moreover, animal genotyping will improve not only the correct parental assignment but also will provide with a clearer description of relationships among individuals as well as similarities among different breeds at the genomic level [21]. The use of multiple ovulation embryo transfer (MOET) is another strategy that could be utilized to keep favorable genetic material in the population. Scientific support could be further enhanced via the development of research nucleus per breed to investigate the variability in a set of phenotypes and the potential of selective breeding. For local breeds, historical bonds with the culture can be found and should not be overlooked. To complement this, the development of appropriate marketing of the final product (mainly meat in our case) that will guarantee reliance from the consumers as well as a satisfying income to the farmers should be considered. A techno-economic analysis could be a further step to assess the applicability of different scenarios and to quantify cost–benefit. 

## 5. Conclusions

Our analysis outlined the critical situation, in terms of population size and genetic diversity, for the Pontremolese, Calvana, and Mucca Pisana breeds. Concerning the breeds of Sardinia (Sarda, Sardo Bruna, and Sardo Modicana) low pedigree quality poses restrictions for an accurate assessment of the genetic diversity. However, trends of pedigree completeness in recent years are encouraging and towards the desired direction. Genomic data, favorably of a large and representative sample per breed, are expected to bridge this gap and to shed more light on the genomic background of the aforementioned breeds and the level of their genomic diversity.

As expected, based on the history of the CHA and LIM breeds, there is space for intense selection. However, this should not be thoughtlessly applied, but under an optimized scheme taking into account both an increased genetic gain of the traits under selection and maintenance of genetic diversity. 

## Figures and Tables

**Figure 1 animals-09-00880-f001:**
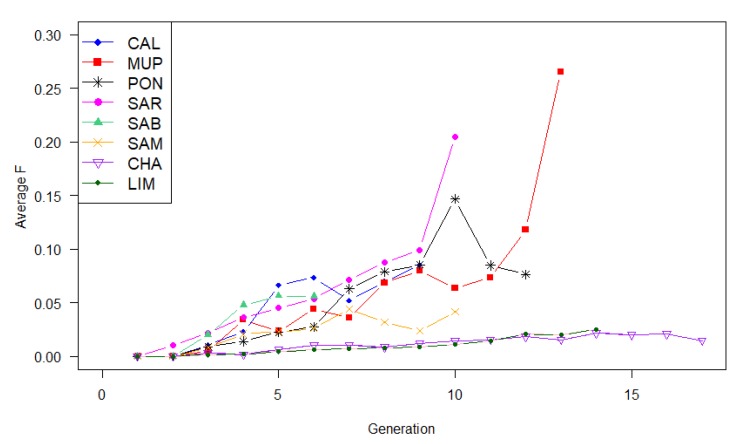
Rate of inbreeding (Δ*F*) per generation for each breed, where CAL = Calvana; MUP = Mucca Pisana; PON = Pontremolese; SAR = Sarda; SAB = Sardo Bruna; SAM = Sardo Modicana; CHA = Charolais; LIM = Limousine.

**Figure 2 animals-09-00880-f002:**
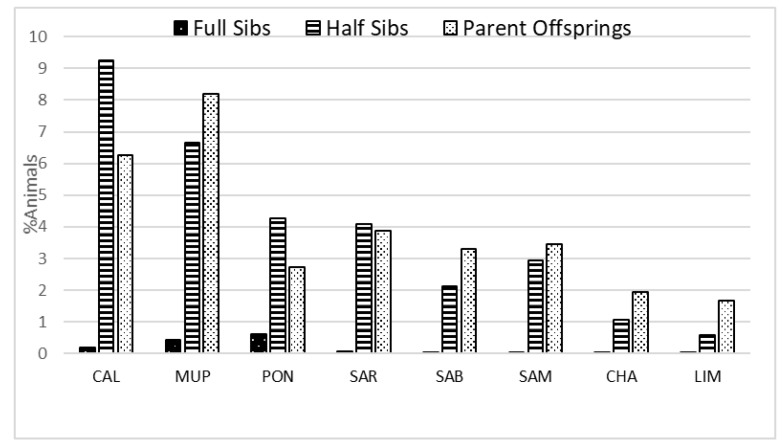
Percentage of animals involved in matings between close relatives (between full-siblings (sibs), half-sibs, and parent–offspring) in each breed, where CAL = Calvana; MUP = Mucca Pisana; PON = Pontremolese; SAR = Sarda; SAB = Sardo Bruna; SAM = Sardo Modicana; CHA = Charolais; LIM = Limousine.

**Figure 3 animals-09-00880-f003:**
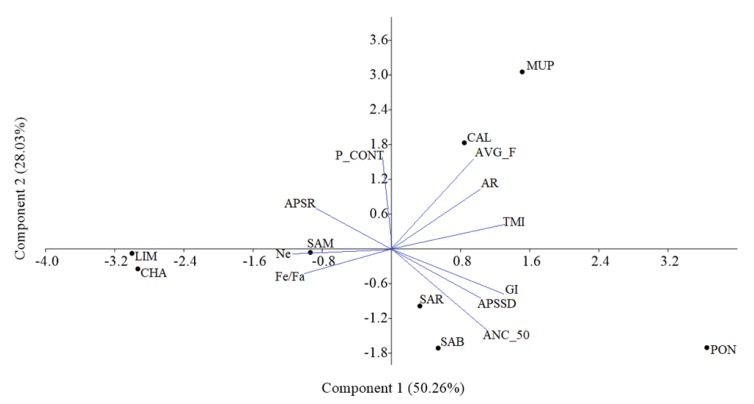
Biplot of the first two principal components. Principal component analysis (PCA) performed on the following population parameters: true mean inbreeding (TMI); average coefficient inbreeding (AVG_F); average relatedness (AR); effective population size (*Ne*); effective number of founders/effective number of ancestors (*fe*/*fa*); ancestors explaining 50% (ANC_50); pedigree content (P_CONT); population size expressed as average ratio throughout the years (APSR); population size expressed as average standard deviation throughout the years (APSSD); generation interval (GI). The vectors represent the variables and the points represent the breeds.

**Table 1 animals-09-00880-t001:** Number of pedigree records (*N*), number of males, females, and generations for each breed.

Breed ^1^	*N*	Males	Females	Generations
Tuscan				
CAL	2798	1201	1597	10
MUP	3399	1447	1952	14
PON	328	147	181	13
Sardinian				
SAR	97,163	28,869	68,294	11
SAB	74,981	13,697	61,284	10
SAM	25,355	10,398	14,957	12
Cosmopolitan				
CHA	99,464	39,171	60,293	18
LIM	322,321	133,445	188,876	15

^1^ CAL = Calvana; MUP = Mucca Pisana; PON = Pontremolese; SAR = Sarda; SAB = Sardo Bruna; SAM = Sardo Modicana; CHA = Charolais; LIM = Limousine.

**Table 2 animals-09-00880-t002:** Pedigree completeness parameters.

Breed ^1^	EquiGen ^2^	FullGen ^3^	MaxGen ^4^	PCI (%) ^5^
Tuscan				
CAL	2.87	2.04	4.44	66
MUP	3.91	2.44	7.55	74
PON	2.10	1.06	4.24	38
Sardinian				
SAR	1.10	0.64	1.89	22
SAB	0.75	0.45	1.20	15
SAM	1.85	1.08	3.18	39
Cosmopolitan				
CHA	2.79	1.51	6.39	50
LIM	3.07	1.79	5.83	59

^1^ CAL = Calvana; MUP = Mucca Pisana; PON = Pontremolese; SAR = Sarda; SAB = Sardo Bruna; SAM = Sardo Modicana; CHA = Charolais; LIM = Limousine. ^2^ equiGen = average values of equivalent complete generations. ^3^ fullGen = average values of full complete generations. ^4^ maxGen = average values of maximum complete generations. ^5^ PCI = pedigree completeness index expressed as a percentage.

**Table 3 animals-09-00880-t003:** Generation interval ^1^ in years for each breed ^2^, calculated in the classical four pathways (standard deviations in parenthesis).

Pathway	CAL	MUP	PON	SAR	SAB	SAM	CHA	LIM
Father to sons	9.55_(13.83)_	9.26_(11.33)_	13.20_(14.01)_	11.22_(14.79)_	6.17_(7.69)_	7.26_(11.61)_	6.35_(7.46)_	7.25_(10.11)_
Father to daughters	10.37_(14.17)_	9.86_(11.04)_	17.12_(17.59)_	10.66_(14.24)_	7.24_(8.70)_	7.35_(11.45)_	5.46_(4.72)_	6.22_(6.89)_
Mother to sons	10.54_(13.30)_	6.82_(7.32)_	8.10_(10.25)_	8.57_(8.47)_	10.86_(12.68)_	7.88_(6.47)_	7.62_(8.80)_	9.13_(12.55)_
Mother to daughters	10.27_(12.77)_	8.20_(9.78)_	9.85_(9.95)_	10.69_(12.68)_	16.49_(18.96)_	8.19_(8.28)_	7.78_(8.11)_	7.52_(8.04)_
Total Interval	10.29_(13.49)_	8.94_(10.38)_	12.51_(13.86)_	10.60_(13.19)_	13.30_(16.72)_	7.80_(9.54)_	6.69_(7.00)_	7.05_(8.13)_

^1^ Generation interval was measured as the average age of parents at the birth of all their progenies; ^2^ CAL = Calvana; MUP = Mucca Pisana; PON = Pontremolese; SAR = Sarda; SAB = Sardo Bruna; SAM = Sardo Modicana; CHA = Charolais; LIM = Limousine.

**Table 4 animals-09-00880-t004:** Genetic variability parameters for each breed.

Breed ^1^	*Ne* ^2^	Δ*F* (%) ^3^	*F* (%) ^4^	TMI (%) ^5^	AR (%) ^6^
Tuscan					
CAL	19.68	2.54	5.10	6.00	6.39
MUP	18.52	2.70	7.25	8.00	10.54
PON	14.62	3.42	3.64	5.60	7.15
Sardinian					
SAR	16.64	3.00	1.90	5.10	0.04
SAB	18.91	2.64	1.23	5.10	0.05
SAM	39.79	1.26	1.66	2.80	0.37
Cosmopolitan					
CHA	90.29	0.55	0.96	1.30	0.20
LIM	132.65	0.37	0.71	0.90	0.20

^1^ CAL = Calvana; MUP = Mucca Pisana; PON = Pontremolese; SAR = Sarda; SAB = Sardo Bruna; SAM = Sardo Modicana; CHA = Charolais; LIM = Limousine. ^2^
*Ne* = effective population size based on equivalent generations. ^3^ Δ*F* = rate of inbreeding. ^4^
*F* = inbreeding coefficient. ^5^ TMI = true mean inbreeding. ^6^ AR = average relatedness.

**Table 5 animals-09-00880-t005:** Population parameters for each breed.

Breed ^1^	*fe/fa* ^2^	ANC_50 ^3^	P_CONT (%) ^4^	APSR ^5^	APSSD ^6^
Tuscan					
CAL	1.1	8	92	1.40	14.34
MUP	1.1	5	94	−1.42	27.57
PON	1.1	5	73	−103.23	454.11
Sardinian					
SAR	1.2	542	61	−4.03	21.85
SAB	1.2	294	48	2.26	20.08
SAM	1.2	96	78	−4.38	9.50
Cosmopolitan					
CHA	3.0	219	77	3.60	7.05
LIM	2.1	330	83	6.04	6.25

^1^ CAL = Calvana; MUP = Mucca Pisana; PON = Pontremolese; SAR = Sarda; SAB = Sardo Bruna; SAM = Sardo Modicana; CHA = Charolais; LIM = Limousine. ^2^
*fe*/*fa =* ratio of effective number of founders to effective number of ancestors. ^3^ ANC_50 = ancestors explaining 50% of the genetic contribution. ^4^ P_CONT = pedigree content. ^5^ APSR = population size expressed as average ratio throughout the years. ^6^ APSSD = population size expressed as average standard deviation throughout the years.

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
