# Peer review of "Population Structure and Genetic Diversity of Italian Beef Breeds as a Tool for Planning Conservation and Selection Strategies"

_animals, 2019, doi:10.3390/ani9110880_

Round 1

Reviewer 1 Report

The paper are very well performed and innovative.

The article deals with the study of structure, genetic diversity and threat levels of local breeds and establishment of conservation strategies. At this point the paper contributes to the basic knowledge needed in any study of this nature.

However, the study advances when using multivariate strategies to verify the degree of contribution of the variables evaluated, as subsidy to conservation.

Is the text clear and easy to read?  Were the results analyzed and 
interpreted correctly?  

The work is innovative because it combines methodologies traditionally used to obtain more complete answers that will allow to delineate more consistent conservation programs.

The text is well described and allows a fluid reading and the discussion is very well conducted. 

Author Response

AU: We thank the reviewers for their review. Please find below our responses to the points raised. All our responses are preceded by “AU” (and in blue). Changes in the manuscript are highlighted in cyan. We hope to find the new version of the manuscript suitable for publication in the “Animals” journal.

Looking forward to hearing from you.

Sincerely,

Maria-Chiara Fabbri

Reviewer 1

The paper are very well performed and innovative.

The article deals with the study of structure, genetic diversity and threat levels of local breeds and establishment of conservation strategies. At this point the paper contributes to the basic knowledge needed in any study of this nature.

However, the study advances when using multivariate strategies to verify the degree of contribution of the variables evaluated, as subsidy to conservation.

The work is innovative because it combines methodologies traditionally used to obtain more complete answers that will allow to delineate more consistent conservation programs.

The text is well described and allows a fluid reading and the discussion is very well conducted.

AU: We would like to thank the reviewer for the review and the comments.

Reviewer 2 Report

The manuscript “ Population structure and genetic diversity of Italian beef breeds as a tool for planning conservation and selection strategies" is well written and has scientific soundness. This manuscript can contribute to effort of conservation of local breeds and knowledge about the genetic diversity. 

I have some minor comments:  

Line 20: Change imposes to impose

I don’t agree this is the main reason for research in biodiversity. In addition, the authors should be carefully state that "the inability of the agricultural sector to meet with the increasing meat consumption”. This sector is able produce enough meat but the technology and other factors preventing its potentials.

Line 45-46:

“Multivariate analysis .. genetic parameters calculated” this sentence does not give any information, if possible, they authors should give the results or further information,

What are genetic parameters the authors wanted to mention? This term might make animal breeders confused. 

The introduction is well written, but check double space in some places.

Methods:

It is might be useful to include the geographical origin of the breeds

Line 131: why the N for PON breed is fewer

Line 202: Remove Table 2 and move the sentence from 202-204 to the footnote

Do the same for Table 4 and 5, the author should keep Table tittle short, and add the explanation in the footnote

Line 339: change the reference to Gutiérrez et al. [32]and Cañas-Álvarez et al. [33],

Line 339 have analysed to analysed

Line 343; Same results to The similar results

Line 380: Mastrangelo et al. [19] used genomic data for the studied breeds in this paper, I would recommend the authors should make more comparison regarding to the results obtained from two different approaches. In fact, in line 410-411, the authors stated the genomic data is expected to clarify some values …. Are these values can be obtained from results of Mastrangelo et al. [19]

Line 457 the authors should not claim as the first study since it does not makes any scientific sense also these breeds were studied in the Mastrangelo as the text “ In addition, two cosmopolitan breeds reared in Italy (Charolais and Limousin) were included because they are used for cross-breeding with local breeds.”

Author Response

Title: Title Population structure and genetic diversity of Italian beef breeds as a tool for planning conservation and selection strategies

AU: We thank the reviewers for their review. Please find below our responses to the points raised. All our responses are preceded by “AU” (and in blue). Changes in the manuscript are highlighted in cyan. We hope to find the new version of the manuscript suitable for publication in the “Animals” journal.

Looking forward to hearing from you.

Sincerely,

Maria-Chiara Fabbri

The manuscript “Population structure and genetic diversity of Italian beef breeds as a tool for planning conservation and selection strategies" is well written and has scientific soundness. This manuscript can contribute to effort of conservation of local breeds and knowledge about the genetic diversity.

Minor comments

Line 20: Change imposes to impose

AU: Changed as suggested.

I don’t agree this is the main reason for research in biodiversity. In addition, the authors should be carefully state that "the inability of the agricultural sector to meet with the increasing meat consumption”. This sector is able produce enough meat but the technology and other factors preventing its potentials.

AU: We do agree with the reviewer that the topic is debatable. We have rephrased this sentence (L19-21).

Line 45-46:

“Multivariate analysis. genetic parameters calculated” this sentence does not give any information, if possible, they authors should give the results or further information,

What are genetic parameters the authors wanted to mention? This term might make animal

breeders confused.

AU: This part has been deleted.

The introduction is well written but check double space in some places.

AU: Changed as suggested.

Methods:

It is might be useful to include the geographical origin of the breeds

AU: The geographical origin of the breeds, i.e. the regions of Tuscany and Sardinia, is already stated in the manuscript [Simple Summary (L26-30), Summary (L34-38), Introduction (L117-118, 122-125) and in all tables].

Line 131: why the N for PON breed is fewer

AU: Pontremolese is classified by FAO at critical risk of extinction. The full pedigree provided by the Italian Breeders Association (AIA) contained 328 animals (L130). Also, there was a typo in the Table 1 where the number of males and females of PON and MUP were oppositely assigned. This has been corrected now.

Line 202: Remove Table 2 and move the sentence from 202-204 to the footnote

AU: We assumed that the reviewer suggested to reduce the title of Table 2 and not to fully remove it and we made the changes accordingly (L202).

Do the same for Table 4 and 5, the author should keep Table tittle short, and add the explanation in the footnote

AU: Changed as above (L242 and 265).

Line 339: change the reference to Gutiérrez et al. [32]and Cañas-Álvarez et al. [33],

AU: Changed as suggested (L357).

Line 339 have analysed to analysed

AU: Changed as suggested (L357).

Line 343; Same results to The similar results

AU: Changed as suggested (L361).

Line 380: Mastrangelo et al. [19] used genomic data for the studied breeds in this paper, I would recommend the authors should make more comparison regarding to the results obtained from two different approaches. In fact, in line 410-411, the authors stated the genomic data is expected to clarify some values …. Are these values can be obtained from results of Mastrangelo et al. [19]

AU: We see the two approaches as complementary to each other. Pedigree analysis is able to capture all IBD relationships, given a correct pedigree. On the other hand, genomic information can identify IBS relationships, that might be due to random effects, yet increase similarity between individuals in the population. Nevertheless, genomic data should be representative of the population, capturing, if possible, all of the variation. The paper of Mastrangelo et al analyzed a small sample of each breed ~24 animals/breed (now reported in the text in L401-404) most likely derived from a limited number of farms and generations. To alleviate this discrepancy, a future genomic analysis utilizing a large number of animals of each breed might be of much help. Indeed, the current project aims in genotyping ~3k samples of the local breeds, and ~5k of the Charolais and Limousine breeds. For the last two, the genotypes are primarily for the implementation of a single step GBLUP in their genomic breeding scheme. Hence, at this point, we can only report the Ne values derived from both approaches.

Line 457 the authors should not claim as the first study since it does not makes any scientific sense also these breeds were studied in the Mastrangelo as the text “ In addition, two cosmopolitan breeds reared in Italy (Charolais and Limousin) were included because they are used for cross-breeding with local breeds.”

AU: Indeed, our work is the first study utilizing the full pedigree data of the Italian Charolais and Limousine breeds. As it has been stated in the manuscript (L 330), these breeds have recently set up their own breeding schemes in Italy. We would like to keep the statement in the manuscript, which has been rephrased accordingly (L469-471).

Reviewer 3 Report

The manuscript deals with an important and interesting topic, which is about estimating parameters for genetic diversity of Italian beef breeds. With a few breeds playing a major role in the industry, preserving local genetic resources is important for the future. There are numerous parameters estimated and the advantage of some to the other is not clear. Most of those results end up in the supplement. Parameters such as ANC_50 has no clear justification, and it is not known whether it has been used in other studies. The manuscript is not well written and presented. Some sentences are unnecessarily very long. I invite the authors to revise and resubmit the manuscript, especially in terms of the clearity of presentation, flow of the information, and English language. Please see further comments below:

L20: "imposes the research" -> "imposes the research"
L28: "that are in large numbers but of low productivity. In addition, the study investigated" -> "that are larger in number but of lower productivity, and"
L29: Change to "the cosmopolitan beef breeds Charolais and Limousine."
L31: "characterized the alarming situation of those breeds" -> "is an alarming situation for these breeds"
L32-33: "could benefit from the application of breeding programs following the paradigm of 32 Charolais and Limousine." It is not clear what you mean by that.
L34: "investigated" -> "investigate"
L34: "three" -> "three local"
L34: "eight Italian beef breeds" CHA and LIM are not necessarily Italian.
L37: "(SAM) and" -> "(SAM), and"
L38-39: "greater values were observed for CHA (90.29) and LIM (135.65)" -> "90.29 for CHA, and 135.65 for LIM"
L41: SAM is missing in the parentheses.
L42: Delete "The rates of inbreeding in local breeds ranged from 3.42% (PON) to 1.26% (SAM) and <1% in cosmopolitan breeds."
L43: "For CAL and MUP higher rates of matings between half sibs were observed" -> "The highest rates of matings between half sibs were observed for CAL and MUP"
L44: "while parent-offspring" -> "while the highest rates of parent-offspring"
L44-45: Delete "Multivariate analysis was performed to better understand the relationship between breeds and genetic parameters calculated." These details are not necessary in the abstract. Please add a concluding line instead.
L46: "support" -> "supported"
Keywords: You may add more keywords such as conservation
L50: "9th" -> "the 9th"
L54: "the19th" -> "the 19th"
L55: Rephrase "rose the request"
L55: "leads" -> "led"
L61: "of other local unimproved breeds" -> "in local breeds"
L61: "a valuable" -> "valuable"
L62: "resource" -> "resources"
L63: "following decades" -> "next decades"
L66: "Risk Status of animal Genetic Resources, updated in February 2019" should be cited as a reference, not like this.
L68-69: "In Italy, there have been registered 61 cattle breeds" to "In Italy, 61 cattle breeds have been registered"
L69: "do not exist anymore" -> "are extinct,"
L70-71: "Domestic Animal Diversity Information System, DAD-IS, http://www.fao.org/dad-is/en/" should be cited as a reference, not like this.
L72: "Italiana Allevatori" -> "Italiana Allevatori, CITY?"
L73: "drastic demographic contraction" It is not clear.
L74: "Sardo Bruna and Sardo Modicana" -> "Sardo Bruna, and Sardo Modicana"
L77: "measures" -> "measures for them"
L77: "those" -> "these"
L78: "concrete risk to be extinct" -> "high risk of extinction"
L80: "beef cattle soared" Unclear
L80: "the last decades" -> "in the last decades"
L81: This line is unclear.
L82: "are multiple" Unclear
L87: "being" -> "are"
L88: "counterparts, represent" -> "counterparts to their local environment. They represent"
L92: "within" -> "within groups of"
L95-97: Unclear
L97: "Inbreeding and rate of inbreeding per generation (or per year) have" -> "Inbreeding rate (per year or generation) has"
L99: (ΔF) should appear the first time you use "rate of inbreeding" or "inbreeding rate"
L99: "increase of inbreeding" -> "increase in inbreeding"
L100: "level providing" -> "level, providing"
L101: "species because can cause a" -> "species, because it can cause"
L102: "decrease in performance" Which performance or performances?
L102: "a reduction of fitness" -> "reduced fitness"
L102: "further" -> "also"
L104: "participates" -> "participate"
L105: "individuals" -> "the total number of individuals"
L105: "factor which" -> "factor, which"
L105: "understand the trend of" -> "understanding the"
L106: "the conservation status" -> "its conservation status"
L108: Delete "defines the different"
L108: "changes between generations" Which changes? Please be specific.
L109: "for the whole" -> "in the"
L111: "selection but" -> "selection, but"
L113: "population and" -> "population, and"
L116: "variability" -> "diversity"
L120: "breeds while" -> "breeds, while"
L122: "some" -> "a few"?
L122: "per breed" -> "for each of these two breeds"
L122-125: This sentence is too long.
Materials and Methods: It is not explained well. There are several unclear parts. There is no time interval for the data.
L129: "Italiana Allevatori" -> "Italiana Allevatori, CITY?"
L130: "LIM and" -> "LIM, and"
L130: "summarized" -> "summarizes"
L131: "Number of full pedigree records" -> "Number of pedigree records"
Table 1: This table is incorrect! Males + Females != N
Table 1: Indent breed abbreviations
Table 1: "Tuscan" -> "Tuscanian"
Table 1 & elsewhere: Put spaces around "="
L135: "sum for (1/2)n" n is the number of generation. What is it summing up?
L136: "was the number of generations separating the individual" -> "is the number of generations separating individuals"
L137: "generations" -> "number of generations"
L137: The difference between (ii) and (iii) is not clear.
L139: "with all ancestors known" It is impossible! At some point there would be unknown ancestors.
L142: "contributions" which contributions?
L146-147: "i and d is the number of generations that are taken into account." -> "i, and d is the total number of generations."
L148: "as the average" -> "i) the average"
L149: "or only for those used for reproduction" -> "ii) the average age of parents at the birth of the progenies that were used for reproduction"
L154: "Meuwissen and Luo (1992)" Incorrect citation
L157: "Ne was the effective population size" -> "Ne is the effective population size"
L158: "regression as following" -> "the following"
L160: "over the equivalent complete generations" Unclear
L161: "equiGen" This abbreviation has not been introduced before.
L163 and elsewhere: " sibs" -> "-sibs"
L164: "multivariate analysis" - "principle component analysis (PCA)"
L166: "reported by Boichard" -> "Boichard"
L167: "and effective number of ancestors" -> "and the effective number of ancestors"
L169: "into account the" -> "into account respectively the"
L169: "as percentage on reference population" How? and how do you define the rreference population?
L170: How do you define "bottleneck" and "the founder effect in the populations"?
L171: "multivariate analysis" -> "PCA analysis"
L171: "relative change of population" What does it mean?
L174: "Pedigree content" is not a good phrase. It does not reflect the pedigree completeness.
L174: "for each generation" -> "in each generation"
L177-178: Delete these lines. The content is already mentioned.
L179: "Multivariate Analysis" -> "PCA Analysis"
L185: "number of ancestor explaining 50%" 50% of what? Why this parameter should be interesting?
L187: "based on equivalent generation" Unclear
L189-195: Please re-write or delete. These lines are unclear.
L196: "graph was developed" -> "graphs were produced"
L198: "Data and Pedigree Analysis" -> "Pedigree Analysis"
L199: Where is SAR?
L200: Give a little explanation about the sex ratio.
Table 2: Indent breed abbreviations
Table 2: "Tuscan" -> "Tuscanian"
Table 2: fullGen and maxGen are not introduced in materials and methods.
L208: "Intermediate values" -> "Intermediate PCI values"
L209: "lowest values" -> "lowest PCI values"
L210: "both indices" Which indices?
There are many more problems and I stop typing here.
Reference [21] has no title!

Author Response

Title: Title Population structure and genetic diversity of Italian beef breeds as a tool for planning conservation and selection strategies

AU: We thank the reviewers for their review. Please find below our responses to the points raised. All our responses are preceded by “AU” (and in blue). Changes in the manuscript are highlighted in cyan. We hope to find the new version of the manuscript suitable for publication in the “Animals” journal.

Looking forward to hearing from you.

Sincerely,

Maria-Chiara Fabbri

Major comments

The manuscript deals with an important and interesting topic, which is about estimating parameters for genetic diversity of Italian beef breeds. With a few breeds playing a major role in the industry, preserving local genetic resources is important for the future. There are numerous parameters estimated and the advantage of some to the other is not clear. Most of those results end up in the supplement. Parameters such as ANC_50 has no clear justification, and it is not known whether it has been used in other studies. The manuscript is not well written and presented. Some sentences are unnecessarily very long. I invite the authors to revise and resubmit the manuscript, especially in terms of the clearity of presentation, flow of the information, and English language. Please see further comments below:

AU: Changes have been made throughout the manuscript to improve quality of the paper (please see the highlighted text in the manuscript). We hope that the new version will meet with the expectations of the reviewer.

As it has been stated in the manuscript (L173), ANC_50 has previously been used in studies such as Bouquet et al., (2011); Cañas-Álvarez et al., (2014); it was also be used by Cavani et al., (2018) and Falleiro et al., (2014).

Bouquet, A., Venot, E., Laloë, D., Forabosco, F., Fogh, A., Pabiou, T., Moore, K., Eriksson, J.-Å., Renand, G., Phocas, F., 2011. Genetic structure of the European Charolais and Limousin cattle metapopulations using pedigree analyses. J. Anim. Sci. 89, 1719–1730. https://doi.org/10.2527/jas.2010-3469

Cañas-Álvarez, J.J., Gónzalez-Rodríguez, A., Martín-Collado, D., Avilés, C., Altarriba, J., Baro, J.A., De la Fuente, L.F., Díaz, C., Molina, A., Varona, L., Piedrafita, J., 2014. Monitoring changes in the demographic and genealogical structure of the main Spanish local beef breeds. J. Anim. Sci. 92, 4364–4374. https://doi.org/10.2527/jas.2013-7420

Cavani, L., Silva, R.M. de O., Carreño, L.O.D., Ono, R.K., Bertipaglia, T.S., Farah, M.M., Millen, D.D., Fonseca, R. da, 2018. Genetic diversity of Brazilian Brahman cattle by pedigree analysis. Pesqui. Agropecuária Bras. 53, 74–79. https://doi.org/10.1590/s0100-204x2018000100008

Falleiro, V., Malhado, C., Malhado, A.C., Carneiro, P., Carrillo, J., Song, J., 2014. Population Structure and Genetic Variability of Angus and Nellore Herds. J. Agric. Sci. 6, p276. https://doi.org/10.5539/jas.v6n12p276

Minor comments

L20: "imposes the research" -> "impose the research"

AU: Changed as suggested.

L28: "that are in large numbers but of low productivity. In addition, the study investigated" -> "that are larger in number but of lower productivity, and"

AU: Changed as suggested (L28).

L29: Change to "the cosmopolitan beef breeds Charolais and Limousine."

AU: Changed as suggested (L29).

L31: "characterized the alarming situation of those breeds" -> "is an alarming situation for these breeds"

AU: Changed as suggested (L31).

L32-33: "could benefit from the application of breeding programs following the paradigm of 32 Charolais and Limousine." It is not clear what you mean by that.

AU: The statement has been rephrased (L32).

L34: "investigated" -> "investigate"

AU: Changed as suggested (L34).

L34: "three" -> "three local"

AU: Changed as suggested (L34).

L34: "eight Italian beef breeds" CHA and LIM are not necessarily Italian.

AU: Changed as suggested (L34). Indeed, however, as it has been stated in the manuscript (L 329-330), these breeds have recently set up their own breeding schemes in Italy.

L37: "(SAM) and" -> "(SAM), and"

AU: Changed as suggested (L37).

L38-39: "greater values were observed for CHA (90.29) and LIM (135.65)" -> "90.29 for CHA, and 135.65 for LIM"

AU: Changed as suggested (L39).

L41: SAM is missing in the parentheses.

AU: Now it has been included (L41).

L42: Delete "The rates of inbreeding in local breeds ranged from 3.42% (PON) to 1.26% (SAM) and <1% in cosmopolitan breeds."

AU: Changed as suggested.

L43: "For CAL and MUP higher rates of matings between half sibs were observed" -> "The highest rates of matings between half sibs were observed for CAL and MUP"

AU: Changed as suggested (L42).

L44: "while parent-offspring" -> "while the highest rates of parent-offspring"

AU: Changed as suggested (L43).

L44-45: Delete "Multivariate analysis was performed to better understand the relationship between breeds and genetic parameters calculated." These details are not necessary in the abstract. Please add a concluding line instead.

AU: This sentence has been deleted.

L46: "support" -> "supported"

AU: We would like to keep a present tense for the conclusions (L44-45).

Keywords: You may add more keywords such as conservation

AU: Changed as suggested (L47).

L50: "9th" -> "the 9th"

AU: Changed as suggested (L49).

L54: "the19th" -> "the 19th"

AU: Changed as suggested (L53).

L55: Rephrase "rose the request"

AU: Changed as suggested (L54).

L55: "leads" -> "led"

AU: Changed as suggested (L55).

L61: "of other local unimproved breeds" -> "in local breeds"

AU: Changed as suggested (L60).

L61: "a valuable" -> "valuable"

AU: Changed as suggested (L60).

L62: "resource" -> "resources"

AU: Changed as suggested (L60).

L63: "following decades" -> "next decades"

AU: Changed as suggested (L62).

L66: "Risk Status of animal Genetic Resources, updated in February 2019" should be cited as a reference, not like this.

AU: Changed as suggested (L64).

L68-69: "In Italy, there have been registered 61 cattle breeds" to "In Italy, 61 cattle breeds have been registered"

AU: Changed as suggested (L66-67).

L69: "do not exist anymore" -> "are extinct,"

AU: Changed as suggested (L67).

L70-71: "Domestic Animal Diversity Information System, DAD-IS, http://www.fao.org/dad-is/en/" should be cited as a reference, not like this.

AU: Changed as suggested (L68-69).

L72: "Italiana Allevatori" -> "Italiana Allevatori, CITY?"

AU: Changed as suggested (L69).

L73: "drastic demographic contraction" It is not clear.

AU: It has been rephrased to “at risk of extinction” (L70).

L74: "Sardo Bruna and Sardo Modicana" -> "Sardo Bruna, and Sardo Modicana"

AU: Changed as suggested (L71).

L77: "measures" -> "measures for them"

AU: Changed as suggested (L74).

L77: "those" -> "these"

AU: Changed as suggested (L75).

L78: "concrete risk to be extinct" -> "high risk of extinction"

AU: Changed as suggested (L75-76).

L80: "beef cattle soared" Unclear

AU: The sentence has been rephrased (L78-80).

L80: "the last decades" -> "in the last decades"

AU: Changed as suggested (L78).

L81: This line is unclear.

AU: The sentence has been rephrased (L79-80).

L82: "are multiple" Unclear

AU: The sentence has been rephrased (L79).

L87: "being" -> "are"

AU: Changed as suggested (L84).

L88: "counterparts, represent" -> "counterparts to their local environment. They represent"

AU: Changed as suggested (L85).

L92: "within" -> "within groups of"

AU: Changed as suggested (L90).

L95-97: Unclear

AU: The sentence has been rephrased (L93-95).

L97: "Inbreeding and rate of inbreeding per generation (or per year) have" -> "Inbreeding rate (per year or generation) has"

AU: Changed as suggested (L95).

L99: (ΔF) should appear the first time you use "rate of inbreeding" or "inbreeding rate"

AU: Changed as suggested (L97).

L99: "increase of inbreeding" -> "increase in inbreeding"

AU: Changed as suggested (L97).

L100: "level providing" -> "level, providing"

AU: Changed as suggested (L98).

L101: "species because can cause a" -> "species, because it can cause"

AU: Changed as suggested (L99).

L102: "decrease in performance" Which performance or performances?

AU: Explanation has been added in the manuscript (L100).

L102: "a reduction of fitness" -> "reduced fitness"

AU: Changed as suggested (L101).

L102: "further" -> "also"

AU: Changed as suggested (L101).

L104: "participates" -> "participate"

AU: Changed as suggested (L102).

L105: "individuals" -> "the total number of individuals"

AU: Changed as suggested (L103).

L105: "factor which" -> "factor, which"

AU: Changed as suggested (L104).

L105: "understand the trend of" -> "understanding the"

AU: Changed as suggested (L104).

L106: "the conservation status" -> "its conservation status"

AU: Changed as suggested (L105).

L108: Delete "defines the different"

AU: Changed as suggested (L106).

L108: "changes between generations" Which changes? Please be specific.

AU: The sentence has been rephrased (L107).

L109: "for the whole" -> "in the"

AU: Changed as suggested (L108).

L111: "selection but" -> "selection, but"

AU: Changed as suggested (L110).

L113: "population and" -> "population, and"

AU: Changed as suggested (L112).

L116: "variability" -> "diversity"

AU: Changed as suggested (L115).

L120: "breeds while" -> "breeds, while"

AU: Changed as suggested (L119).

L122: "some" -> "a few"?

AU: Changed as suggested (L120).

L122: "per breed" -> "for each of these two breeds"

AU: Changed as suggested (L121).

L122-125: This sentence is too long.

AU: The sentence has been rephrased (L121-124).

Materials and Methods: It is not explained well. There are several unclear parts. There is no time interval for the data.

AU: The Material and Methods are presented in a logical order, i.e. Data, Pedigree, Genetic and Principal component analysis. Moreover, changes have been done in this section that we hope to clarify the parts that were unclear for the reviewer. Concerning the data interval, we have used the complete and most recent information registered in the pedigree of each breed since the creation of the different Herdbooks (1980-2018). The data have been supplied by AIA (for the local breeds) and by the breeding association (ANACLI) for the two cosmopolitan breeds (Charolais and Limousine) (L127-130).

L129: "Italiana Allevatori" -> "Italiana Allevatori, CITY?"

AU: Changed as suggested (L128).

L130: "LIM and" -> "LIM, and"

AU: Changed as suggested (L131).

L130: "summarized" -> "summarizes"

AU: Changed as suggested (L131).

L131: "Number of full pedigree records" -> "Number of pedigree records"

AU: Changed as suggested (L134).

Table 1: This table is incorrect! Males + Females != N

AU: Corrections have been done in Table 1.

Table 1: Indent breed abbreviations

AU: Breed abbreviations have been included as a footnote in the tables such that each table is self-explanatory.

Table 1: "Tuscan" -> "Tuscanian"

AU: The term “Tuscan” is correct and has been previously used (Bozzi et al., 2012; Cecchi et al., 2012).

Bozzi, R., Alvarez, I., Crovetti, A., Fernández, I., De Petris, D., Goyache, F., 2012. Assessing priorities for conservation in Tuscan cattle breeds using microsatellites. Anim. Int. J. Anim. Biosci. 6, 203–211. https://doi.org/10.1017/S1751731111001443

Cecchi, F., Ciampolini, R., Castellana, E., Ciani, E., 2012. Genetic diversity within and among endangered local cattle breeds from Tuscany ( Italy ). Large Animal Review, pp. 79–85.

Table 1 & elsewhere: Put spaces around "="

AU: Changed as suggested.

L135: "sum for (1/2)n" n is the number of generation. What is it summing up?

AU: the sum is: ∑ (1/2) elevated to the number of generations identified per each individual studied. The definition has been given by Gutiérrez and Goyache, (2005). The sentence has also been rephrased (L140-143)

Gutiérrez, J.P., Goyache, F., 2005. A note on ENDOG: a computer program for analysing pedigree information. J. Anim. Breed. Genet. 122, 172–176. https://doi.org/10.1111/j.1439-0388.2005.00512.x

L136: "was the number of generations separating the individual" -> "is the number of generations separating individuals"

AU: Changed as suggested (L140).

L137: "generations" -> "number of generations"

AU: Changed as suggested (L140).

L137: The difference between (ii) and (iii) is not clear.

AU: The parameter full complete generations is defined as those generations separating the offspring of the furthest generation where both lines of the individual are known. Ancestors with no known parent are considered as founders. The maximum complete generations is defined as the number of generations separating the individual from its furthest ancestor (Gutiérrez and Goyache, (2005). Changes have also been made in the manuscript (L141-144).

Gutiérrez, J.P., Goyache, F., 2005. A note on ENDOG: a computer program for analysing pedigree information. J. Anim. Breed. Genet. 122, 172–176. https://doi.org/10.1111/j.1439-0388.2005.00512.x

L139: "with all ancestors known" It is impossible! At some point there would be unknown ancestors.

AU: At the point that unknown ancestors appear in the pedigree we stop (actually, 1 generation before, where both ancestors are known). This is the definition of the “full complete generations”: How many generations back in the pedigree we can go for an individual till 1 ancestor will be unknown.

L142: "contributions" which contributions?

AU: Contribution is the average proportion of known ancestors calculated separately for the maternal and paternal lines.  This part has been rephrased (L149).

L146-147: "i and d is the number of generations that are taken into account." -> "i, and d is the total number of generations."

AU: Changed as suggested (L154).

L148: "as the average" -> "i) the average"

AU: Changed as suggested (L155).

L149: "or only for those used for reproduction" -> "ii) the average age of parents at the birth of the progenies that were used for reproduction"

AU: Changed as suggested (L156).

L154: "Meuwissen and Luo (1992)" Incorrect citation

AU: Changed as suggested (L162).

L157: "Ne was the effective population size" -> "Ne is the effective population size"

AU: Changed as suggested (L165).

L158: "regression as following" -> "the following"

AU: We would like to keep the original version of the sentence (L166)

L160: "over the equivalent complete generations" Unclear

AU:  A regression was fit with inbreeding coefficients as dependent variable and equivalent complete generations as the independent variable. The regression coefficient of this model was used in the formula in line 167.

L161: "equiGen" This abbreviation has not been introduced before.

AU: The term is introduced now in L139.

L163 and elsewhere: " sibs" -> "-sibs"

AU: Changed as suggested throughout the manuscript (e.g., L171, L287).

L164: "multivariate analysis" - "principle component analysis (PCA)"

AU: This part has been deleted.

L166: "reported by Boichard" -> "Boichard"

AU: The sentence has been rephrased (L176).

L167: "and effective number of ancestors" -> "and the effective number of ancestors"

AU: The sentence has been rephrased (L190).

L169: "into account the" -> "into account respectively the"

AU: The sentence has been rephrased (L172).

L169: "as percentage on reference population" How? and how do you define the rreference population?

AU: This part has been rephrased (L174). Reference population was defined according to Gutiérrez and Goyache (2005)

Gutiérrez, J.P., Goyache, F., 2005. A note on ENDOG: a computer program for analysing pedigree information. J. Anim. Breed. Genet. 122, 172–176. https://doi.org/10.1111/j.1439-0388.2005.00512.x

L170: How do you define "bottleneck" and "the founder effect in the populations"?

AU: Population bottleneck occurs when a population's size is strongly reduced for at least one generation. A founder effect occurs when a new colony is started by a few members of the original population.

L171: "multivariate analysis" -> "PCA analysis"

AU: The sentence has been deleted.

L171: "relative change of population" What does it mean?

AU: This part has been rephrased “relative change of population size” (L191-192)

L174: "Pedigree content" is not a good phrase. It does not reflect the pedigree completeness.

AU: The term “Pedigree content” is used by Gutiérrez and Goyache, (2005) in their guideline to use ENDOG 4.8 software, hence we would like to keep this term.

Gutiérrez, J.P., Goyache, F., 2005. A note on ENDOG: a computer program for analysing pedigree information. J. Anim. Breed. Genet. 122, 172–176. https://doi.org/10.1111/j.1439-0388.2005.00512.x

L174: "for each generation" -> "in each generation"

AU: The sentence has been rephrased (L177).

L177-178: Delete these lines. The content is already mentioned.

AU: Changed as suggested.

L179: "Multivariate Analysis" -> "PCA Analysis"

AU: Changed as suggested (L181).

L185: "number of ancestor explaining 50%" 50% of what? Why this parameter should be interesting?

AU: This parameter is provided by the software and has been widely used in the literature. It is defined as the number of ancestors explaining 50% of the genetic contribution in the population (L174). This parameter has been used by many authors, such as Bouquet et al., (2011); Cañas-Álvarez et al., (2014); Cavani et al., (2018); Falleiro et al., (2014), and it can indicate the presence of the founder effect, increasing the knowledge of population structure of the breeds under study.

Bouquet, A., Venot, E., Laloë, D., Forabosco, F., Fogh, A., Pabiou, T., Moore, K., Eriksson, J.-Å., Renand, G., Phocas, F., 2011. Genetic structure of the European Charolais and Limousin cattle metapopulations using pedigree analyses. J. Anim. Sci. 89, 1719–1730. https://doi.org/10.2527/jas.2010-3469

Cañas-Álvarez, J.J., Gónzalez-Rodríguez, A., Martín-Collado, D., Avilés, C., Altarriba, J., Baro, J.A., De la Fuente, L.F., Díaz, C., Molina, A., Varona, L., Piedrafita, J., 2014. Monitoring changes in the demographic and genealogical structure of the main Spanish local beef breeds. J. Anim. Sci. 92, 4364–4374. https://doi.org/10.2527/jas.2013-7420

Cavani, L., Silva, R.M. de O., Carreño, L.O.D., Ono, R.K., Bertipaglia, T.S., Farah, M.M., Millen, D.D., Fonseca, R. da, 2018. Genetic diversity of Brazilian Brahman cattle by pedigree analysis. Pesqui. Agropecuária Bras. 53, 74–79. https://doi.org/10.1590/s0100-204x2018000100008

Falleiro, V., Malhado, C., Malhado, A.C., Carneiro, P., Carrillo, J., Song, J., 2014. Population Structure and Genetic Variability of Angus and Nellore Herds. J. Agric. Sci. 6, p276. https://doi.org/10.5539/jas.v6n12p276

L187: "based on equivalent generation" Unclear

AU: The effective population size has been calculated in 3 different ways as mentioned in M&M (L139-144). For the PCA, the Ne that has been calculated based on equivalent generations was used (L191).

L189-195: Please re-write or delete. These lines are unclear.

AU: This part has been deleted.

L196: "graph was developed" -> "graphs were produced"

AU: This part has been deleted (L194).

L198: "Data and Pedigree Analysis" -> "Pedigree Analysis"

AU: Changed as suggested (L196).

L199: Where is SAR?

AU: SAR has been added (L197).

L200: Give a little explanation about the sex ratio.

AU: Sex ratio is the ratio between Males and Females for each generation (L197). Explanation has given in the text.

Table 2: Indent breed abbreviations

AU: Breed abbreviations have been included as a footnote in the tables such that each table is self-explanatory.

Table 2: "Tuscan" -> "Tuscanian"

AU: The term “Tuscan” is correct and has been previously used (Bozzi et al., 2012; Cecchi et al., 2012).

Bozzi, R., Alvarez, I., Crovetti, A., Fernández, I., De Petris, D., Goyache, F., 2012. Assessing priorities for conservation in Tuscan cattle breeds using microsatellites. Anim. Int. J. Anim. Biosci. 6, 203–211. https://doi.org/10.1017/S1751731111001443

Cecchi, F., Ciampolini, R., Castellana, E., Ciani, E., 2012. Genetic diversity within and among endangered local cattle breeds from Tuscany ( Italy ). Large Animal Review, pp. 79–85.

Table 2: fullGen and maxGen are not introduced in materials and methods.

AU: Both terms have now been abbreviated in M&M (L141-142).

L208: "Intermediate values" -> "Intermediate PCI values"

AU: Changed as suggested (L210).

L209: "lowest values" -> "lowest PCI values"

AU: Changed as suggested (L211).

L210: "both indices" Which indices?

AU: The sentence has been rephrased (L211-212).

There are many more problems and I stop typing here.

AU: We have been throughout the manuscript and changes have been made. We would like again the reviewer for his review. We hope that the new version will meet with the expectations of the reviewer.

Reference [21] has no title!

AU: Title has been added to the reference 21 (now reference 23).

Round 2

Reviewer 3 Report

A major improvement has been made. Now, the manuscript has a better flow and it is readable. Still, the data time interval needs to be addressed. If it is different for different breeds, it needs to be addressed too. The "Results" section is lengthy and most of it is repeating the numbers in the Table. Indicating numbers related to the Figures, in the text is OK. Instead of repeating the numbers that are already in the Table, give an overview about the situation. Still there are many issues, especially in Results and Discussions that require revision. Please see below for further comments.

L41: (1.23, 1.66 and 1.90% for SAB, SAM and SAR, respectively)
L68: "in a vulnerable situation" or "in vulnerable situations"
L79: "a series of advantages" to "advantages"
L84: "breeds, are" to "breeds are"
L88: Rephrase "for a wide use by farmers".
L96: "evolved" to "evolves"
L104: "which contributes understanding the trend of" to "that contributes to the"
L109: "and rate" to "and the rate"
L112: "relationship" to "relationships"
L122: "Italian Limousine (LIM) and Charolais (CHA) populations" to "Italian populations of Limousine (LIM) and Charolais (CHA)"
L123-124: Delete these lines.
Table 1: Please indent breed abbreviations (i.e., type 2-3 spaces before the abbreviation in the table). The same for other tables.
L147: sire and dam as subscript
L143: "number of generations separating the offspring from the furthest generation, where both lines of the individual are"
L143: "both lines" to "both parents"?
L168: "the equivalent complete generations" to "equiGen"
L169: "incompleteness the" to "incompleteness, the"
L175: "Values equal to 1 for fe/fa ratio, show" to "Ratio fe/fa = 1 shows"
L175: "while" to "and"?
L176: "founder effect" to "founder influence on genetic contribution"?
L190: "generation interval (GI)" to "GI"
L192-193: How APSR and APSSD are calculated?
L193: "throughout the years" to "throughout the years (YYYY - YYYY)"
L211: "values, 22% and 15%, respectively" to "values (22% and 15%, respectively)"
Figures: I suggest using continuous lines rather than broken lines.
L213: Continue with the previous paragraph rather than many short paragraphs.
L213: "generation interval (GI)" to "GI". Please do not introduce the same abbreviation several time. Introduce it in the first instance and then continue using the abbreviation.
L215-216: "As expected, LIM and CHA had on average lower GI (6.7 and 7.0 for CHA for LIM, respectively), as showed in Table 3." to "As showed in Table 3, LIM and CHA had lower average GI (7.0 and 6.7, respectively)."
L218-219: "pathways. In parenthesis the standard deviations." to "pathways (standard deviations in parenthesis)."
L218: "breed" to "breed1"
L221: "CAL" to "1 CAL"
L224-226: These line do not provide important information. Please delete them.
L238-239: "Sardinian breeds showed lower values ranging between 1.23% (SAB) to 1.90% (SAR). SAM had an intermediate value (1.66%)" to "Sardinian breeds showed lower values (1.23, 1.66 and 1.90% for SAB, SAM and SAR, respectively)"
L249-250: "In general, inbreeding increased by generation. For CHA and LIM the increase was relatively small (Figure 1)."
L255-263: Here and elsewhere, information given in tables/figures are repeated again in the text. Please do not repeat what you have already presented in the tables/figures, in the text. Just give an overview about the results in the table/figure.
L273-279: Repeated information from Table 5. With repeated numbers in the text, reader tends to skip the text!
Table 5: Because it is not clear how APSR and APSSD are calculated, it is difficult to make sense of the numbers. Please do not expect readers to read other literature to understand your paper. What a positive and negative APSR mean? What causes APSR and APSSD to be so low and high for PON? How these numbers are comparable?
L281: "Limousine breed" to "LIM"
Figure 2: Interesting Figure! I was wondering if you consider the last 4 or 5 generations, does it look similar?
L307-308: "PC1 explained 50.27% of the variability, even though the P_CONT had a coefficient around 0 on this PC." What is the contrast that you want to show using "while"?
L325: "the two Italian cosmopolitan beef breeds, the Charolais and Limousine" to "the two Italian beef populations of CHA and LIM"
L328: "a group of two" to "two"
L331-335: Use breed abbreviations.
L337: "The level of inbreeding within a breed is closely related" This is unclear. Rephrase.
L342: "This, however, was somehow" to "This was somehow"
L343: "is increasing in time and in more recent years" to "has been increasing over time"
L344-346: These lines are unnecessary. Of course Pedigree parameters are different compared to populations in other countries!
L346: "Torrecillas et al. [38]" data on which years?
L346-347: "Mucca Pisana breed" to "MUP"
L351: "Slovenian Limousine" to "Slovenian LIM" I see similar thing in many other places. Why do you have abbreviations if you don't use them?
L350-367: Delete these lines. Due to the nature of this study, comparisons with other breeds and countries is not interesting. There is no surprise if the parameters are different compared to other countries and breeds. Focus on discussing your results, the alarming situations, how these results should be interpreted and used, etc. Comparisons with other Italian breeds can be interesting.
L369-373: The same as the previous comment
L379: "the Ne" to "Ne"
L391: "mark" to "report"
L398: "in here" to "in this study"
L405: "alleviating" to "reducing"?
L406: "upon methodology" to "upon the methodology"
L408: "fitness" is a general term. Revise this sentence.
L425: "Table S1" Why duplicated results?! Table S1 is the exact copy of Figure 2! The averages in Table S1 are useless. You cannot simply average relationships of different kind.
L469: You have mentioned it several times in the manuscript.
I encourage the authors to ask a colleague to read their manuscript and provide them feedback before submitting the manuscript.
